# GCN-DevLSTM: Path Development for Skeleton-Based Action Recognition

**Lei Jiang**                                                                   *lei.j@ucl.ac.uk*
*University College London*

**Weixin Yang**                                                          *weixin.yang@maths.ox.ac.uk*
*University of Oxford*

**Xin Zhang**                                                              *eexinzhang@scut.edu.cn*
*South China University of Technology*

**Hao Ni**                                                                       *h.ni@ucl.ac.uk*
*University College London*

**Reviewed on OpenReview:** *https://openreview.net/forum?id=3o5seglmgn*

## Abstract

Skeleton-based action recognition (SAR) in videos is an important but challenging task in computer vision. The recent state-of-the-art (SOTA) models for SAR are primarily based on graph convolutional neural networks (GCNs), which are powerful in extracting the spatial information from skeleton data. However, their ability to capture temporal dynamics remains limited. To address this, we propose the G-Dev layer, which leverages path development—a principled and parsimonious representation for sequential data based on Lie group structures—to enhance temporal modeling. By integrating the G-Dev layer, the proposed DevLSTM module summarizes local temporal dynamics, reducing the time dimension while retaining high-frequency information. It can be conveniently applied to any temporal graph data, complementing existing advanced GCN-based models. Our empirical studies on the NTU-60, NTU-120 and Chalearn2013 datasets demonstrate that our proposed GCN-DevLSTM network consistently improves the strong GCN baseline models and achieves competitive performance. The code repository is publicly available at https://github.com/DeepIntoStreams/GCN-DevLSTM.

## 1 Introduction

Action recognition has a wide range of applications in diverse fields such as human-computer interaction, performance assessment in athletics, virtual reality, and video monitoring (Wang & Mohamed, 2026). Compared to video-based action recognition, skeleton-based action recognition (SAR) exhibits stronger robustness to varying lighting conditions, improved efficiency in computation and storage (Chen et al., 2021), and better preservation of data privacy. Skeleton data can be obtained either by localization of 2D/3D human joints using depth cameras (Wang & Mohamed, 2026) or pose estimation algorithms from videos (Lal et al., 2025).

Despite extensive research in SAR, designing effective spatio-temporal representations of skeleton sequences remains a key challenge. Previously, Recurrent Neural Networks (RNNs) were commonly used due to their ability to capture temporal dynamics. The advent of Spatial-Temporal Graph Convolutional Networks (STGCNs) (Yan et al., 2018) has shifted the focus towards Graph Convolutional Networks (GCNs), which incorporate both spatial graph convolution and temporal convolution. The majority of subsequent GCN-based approaches (Chen et al., 2021; Shi et al., 2020; Cheng et al., 2020b), following the STGCN framework,

focus on the improvement of the spatial graph convolution, with limited attention to temporal modules. These methods utilize 1-D temporal convolution operations to capture temporal features. However, whether temporal convolution is the most effective approach for modeling complex temporal dynamics remains an open question.

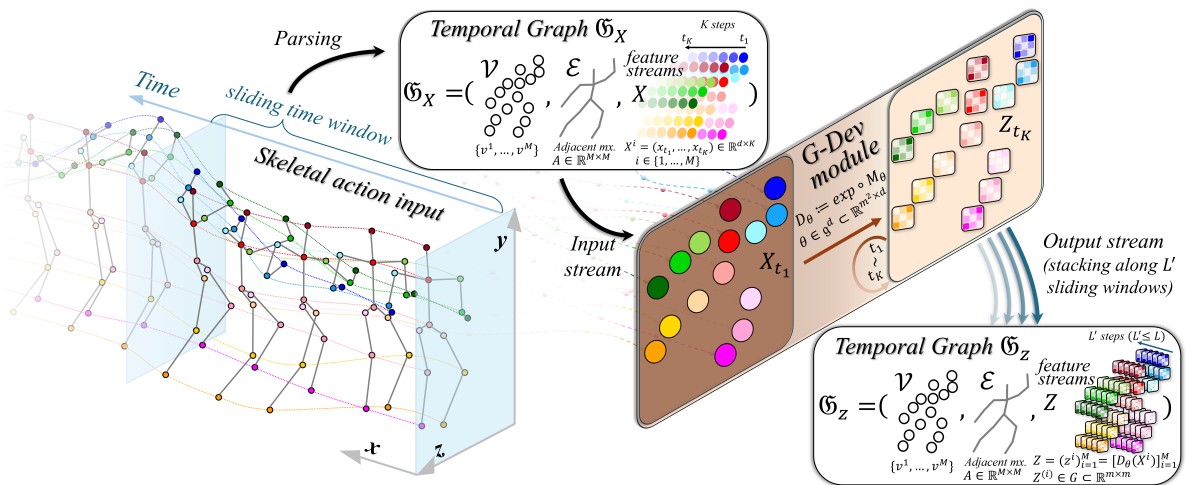

Figure 1: The workflow of G-Dev layer. The G-Dev layer takes a temporal graph (e.g., the skeleton sequence ) as an input and outputs a temporal graph with a potentially significantly reduced time dimension. For each sub-time interval, we apply the G-Dev module (see Section 3.3) to generate a static graph representing the features within that window. We obtain the final temporal graph output by stacking these static graphs along the time dimension.

On the other hand, Rough path theory (Lyons, 1998) originated as a branch of stochastic analysis to make sense of controlled differential equations driven by highly oscillatory paths. More recently, the applications of rough path theory in sequential data analysis are emerging. In particular, the path signature, as a fundamental object of Rough path theory, has demonstrated strong capability in capturing temporal structure across various domains, including SAR (Yang et al., 2022; Shi et al., 2024). By embedding time series into the continuous path space, the signature offers a unified treatment to handle variable length and irregular sampling. However, the path signature may face challenges such as the curse of dimensionality due to high path dimensionality and lack of adaptability due to its deterministic nature (Lou et al., 2024a). To address these, Lou et al. (2024a) propose the path development from rough path theory, as a trainable alternative to path signature. The path development enjoys the desirable properties of the signature while reducing feature dimension significantly. Lou et al. (2024a) show that combining the path development under the appropriate Lie group with LSTM (Hochreiter & Schmidhuber, 1997) alleviates the gradient vanishing, leading to significant enhancement across various sequential tasks.

Motivated by these observations, this paper focuses on improving temporal modeling within GCN-based frameworks. To this end, we introduce a novel G-Dev layer (as shown in Figure 1) on the temporal graph, where each node is associated with time-series features. This layer utilizes a sliding window to summarize local temporal information via the principled path development, offering time dimension benefit and robustness to missing data and irregular sampling. Building on this, we propose the GCN-DevLSTM network, which retains the effectiveness of graph convolution for modeling spatial relationships among joints while leveraging the hybrid capacity of the G-Dev layer and LSTM to capture temporal dynamics. To further enrich structural modeling, we additionally incorporate a line graph stream to capture complementary higher-order relationships between skeletal connections. Our GCN-DevLSTM is flexible and can be integrated with different GCN backbones. Experimental results on NTU-60, NTU-120, and Chalearn2013 demonstrate that our approach consistently improves strong GCN baselines on NTU datasets and achieves competitive performance across all benchmarks. Furthermore, leveraging the properties of path development, our method exhibits significantly enhanced robustness under challenging conditions such as missing or irregular frames.

We summarize our main contributions as follows:

- We design a novel and principled graph development (**G-Dev**) layer for analyzing temporal graph data by extending the path development originally applied to vector-valued time series. As a generic network on graph data with time-varying node attributes, G-Dev exhibits efficacy and robustness to irregular sampling and missing data.

- We introduce the novel **GCN-DevLSTM** network for skeleton-based action recognition that effectively combines the G-Dev layer with LSTM to capture complex temporal dynamics.

- We further incorporate the line graph as an additional stream to capture complementary higher-order structural information, which provides further performance gains in strong multi-stream settings.

- Extensive experiments demonstrate that our method consistently improves strong GCN baselines on NTU datasets and achieves competitive performance across benchmarks, while providing enhanced robustness under challenging conditions.

## 2 Related Work

### 2.1 Skeleton-based Action Recognition

The main challenge in SAR is extracting the effective representation that captures both spatial and temporal dependency of skeleton sequences. Earlier methods typically use convolutional neural networks (CNNs) (Liu et al., 2017b) and RNN (Liu et al., 2017a) to address this challenge without considering structural topology. STGCN (Yan et al., 2018) is the first method to capture spatial and temporal relationships using GCN, taking the skeleton topology of the graph into consideration while this topology is not learnable, with only the spatial connection between adjacent joints being connected. As a result, some actions that rely on non-adjacent joints may lead to poor results. Liu et al. (2020) tackle this issue by introducing a learnable adjacency matrix alongside the original manually defined one, enabling connections between non-neighboring joints. Yet, this approach maintains the same graph for all action inputs, which might not be ideal, as different action classes may benefit from different graph topologies. To address this, Li et al. (2019b); Shi et al. (2019; 2020) propose adaptive GCN with the learnable and data-driven topology that shares among all channels. Cheng et al. (2020a); Chen et al. (2021) further propose the non-shared topology for each channel. To capture more comprehensive spatial dependencies, Kilic et al. (2025); Li et al. (2025) further enhance the skeleton graph by dynamically integrating additional hierarchical structural graphs. Additionally, BlockGCN (Zhou et al., 2024) reveals that purely learnable topologies may suffer from topology degradation, emphasizing the importance of explicitly preserving structural priors. Other recent studies improved model performance by enriching graph representations with higher-order features, such as angle-based information (Schlegel et al., 2024). While these approaches significantly improve spatial modeling, their temporal modeling is typically handled by simple temporal convolution (TCN) modules, which are limited in capturing long-range temporal dependencies. This suggests that temporal modeling remains under-explored in GCN-based frameworks.

Recently, transformer-based methods have been introduced to capture long-range spatial-temporal dependencies via self-attention. For example, LG-STFormer (Liu et al., 2025) and SkateFormer (Do & Kim, 2024) leverage attention mechanisms to model global joint interactions across both spatial and temporal dimensions, and hybrid GCN-transformer architectures (Chen et al., 2025) further combine local graph inductive bias with global attention modeling. Hierarchical transformer-based models such as SkelFormer (Yan et al., 2026) also explore multi-level spatio-temporal representations. However, these methods primarily focus on global dependency modeling through attention mechanisms, rather than improving temporal modeling within graph convolutional frameworks. Therefore, improving temporal modeling within the GCN paradigm remains an important and complementary research direction. In particular, designing more effective temporal modules that can be seamlessly integrated into GCN-based architectures is still underexplored.

## 2.2 Path signature & Path Development

The path signature is a core object in rough path theory that represents a time series through its iterated integrals, capturing both temporal order and interactions across feature dimensions (Lyons, 1998). Its first level records the path increment, while its second level encodes signed area terms; higher levels capture progressively richer geometric structure and higher-order sequential interactions along the path. Taken over all orders, the signature characterizes a path up to negligible time reparametrization, making it a faithful feature representation for sequential data (Levin et al., 2013). This property has enabled a wide range of applications in sequential data learning tasks, including handwritten recognition (Xie et al., 2017), writer identification (Yang et al., 2015), financial data analysis (Lyons et al., 2014), time-series data generation (Ni et al., 2021; Jiang et al., 2025) and SAR (Liao et al., 2021; Cheng et al., 2023; Yang et al., 2022; Ahmad et al., 2019; Li et al., 2019a). Moreover, the signature is invariant to time reparameterization, meaning that traversing the same path at different speeds yields the same representation. This property is particularly desirable in action recognition, where the speed of performing an action should not affect its classification. In the specific context of SAR, Ahmad et al. (2019) focus on employing path signature solely for extracting spatial features while Yang et al. (2022); Li et al. (2019a) utilize path signature to obtain both temporal and spatial features. However, relying solely on path signature to capture spatial relationships may be suboptimal, as it is not data adaptive and only considers local spatial correlations. Liao et al. (2021) leverage GCN to extract spatial features and employ the log signature to capture temporal dynamics. However, such approaches often suffer from the curse of dimensionality, as the feature dimension of the truncated signature grows rapidly with the path dimension, leading to overfitting in deep architectures.

More recently, the path development layer (Lou et al., 2024a) is proposed as a general sequential layer on time series data as a trainable alternative to the signature feature. Instead of computing a fixed set of features, path development maps the input path into a Lie group via a differential equation driven by the path. This results in a compact and learnable representation that retains key advantages of the signature, including sensitivity to temporal ordering and invariance to time reparameterization, while significantly reducing feature dimensionality. The development layer has demonstrated effectiveness in supervised learning on healthcare (Feng et al., 2026), online signature verification (Shi et al., 2025) and synthetic time generation (Lou et al., 2024b). Path development originates from the (Cartan) development of a path (Driver, 1995), and plays an important role in understanding the properties of the path signature (Hambly & Lyons, 2010). In particular, Chevyrev & Lyons (2016) prove that under an appropriate Lie group, path development is a universal and characteristic feature of the path signature, which provides theoretical support for the development layer. Building upon their work (Lou et al., 2024a), we extend the application of path development from vector-valued time series to that of the temporal graph.

## 3 G-Dev Layer: Development layer of a temporal graph

In this section, we start with a concise overview of the development of the path as the principled feature representation of $d$-dimensional time series. Subsequently, we proceed with introducing the path development layer proposed in (Lou et al., 2024a). For details of the path development, we refer readers to (Lou et al., 2024a). Building on this, we propose the G-Dev layer, a novel extension of the development layer applied to temporal graphs, wherein the features of nodes are represented by time series.

### 3.1 The development of a Path

Before the formal definition, we first provide an intuitive explanation of path development. A time series can be viewed as a path evolving over time, and the goal is to construct a representation that captures its temporal dynamics, such as the order of events and how the signal changes, while being robust to variations like speed or sampling rate. Path development achieves this by mapping the input path into a trajectory on a Lie group. Informally, each local segment of the path is associated with a Lie group element, and the path development is obtained by taking the matrix product of these Lie group elements in sequential order. Since this product is order-sensitive, the resulting representation preserves information about the ordering of path segments, thereby encoding the history of the sequence as a matrix-valued representation. The

path development is faithful in the sense that it preserves discriminative information about the shape of the underlying path, making it expressive for distinguishing time series with different geometric and sequential structures. Moreover, its invariance under time reparametrization means that the representation is unaffected by changes in traversal speed, and is therefore robust to variations in sampling rate and temporal resolution.

Let $X \in V([0,T], \mathbb{R}^d)$, where $V([0,T], \mathbb{R}^d)$ represents the space of continuous paths of finite length in $\mathbb{R}^d$ defined on the time interval $[0,T]$. To fix the notations, let $G$ denote a finite-dimensional Lie group and its associate Lie algebra $\mathfrak{g}$. Examples of the matrix Lie algebra associated with the special orthogonal group (denoted by $\mathfrak{so}$) and special Euclidean group ($\mathfrak{se}$) are given as follows:

$$\mathfrak{gl}(m; \mathbb{F}) = \{m \times m \text{ matrices over } \mathbb{F}\} \quad (\mathbb{F} = \mathbb{R} \text{ or } \mathbb{C});$$
$$\mathfrak{so}(m, \mathbb{R}) := \{A \in \mathfrak{gl}(m; \mathbb{R}) : A^T + A = 0\};$$
$$\mathfrak{se}(m) = \left\{ \begin{bmatrix} \omega & v \\ 0 & 0 \end{bmatrix} \mid \omega \in \mathfrak{so}(m, \mathbb{R}), \ v \in \mathbb{R}^m \right\}.$$

**Definition 3.1 (Path Development)** *Let $M : \mathbb{R}^d \to \mathfrak{g} \subset \mathfrak{gl}(m)$ be a linear map. The path development of $X \in V([0,T], \mathbb{R}^d)$ on $G$ under $M$ is the solution $Z_T \in G$ to the below equation*

$$dZ_t = Z_t \cdot M(dX_t) \quad \text{for all } t \in [0,T] \quad \text{with } Z_0 = Id_m, \tag{1}$$

*where $Id_m$ is the identity matrix and $\cdot$ denotes the matrix multiplication.*

**Example 3.1 (Linear path)** *Let $X \in V([0,T], \mathbb{R}^d)$ be a* linear *path. For any linear map $M \in L(\mathbb{R}^d, \mathfrak{g})$, the development of the path under $M$ admits the explicit formula and is simply*

$$D_M(X)_{0,T} = \exp\left(M(X_T - X_0)\right), \tag{2}$$

*where* $\exp$ *is the matrix exponential.*

Thanks to the multiplicative property of the path development (Lemma 2.4 in (Lou et al., 2024a)), the development of two paths $X \in V([0,s], \mathbb{R}^d)$ and $Y \in V([s,t], \mathbb{R}^d)$ under $M$ is the same as the development of the concatenation of these two paths, in formula,

$$D_M(X * Y) = D_M(X) \cdot D_M(Y), \tag{3}$$

where $X * Y$ denotes the concatenation of $X$ and $Y$ and $\cdot$ is the matrix multiplication. Eq. (3) enables us to compute the development of any piecewise linear path explicitly.

The path development enjoys several desirable theoretical properties, making it a principled and efficient representation of sequential data. We summarize the following two properties, that are directly relevant to action recognition.

**Invariance under Time-reparametrization**

Similar to the path signature, path development is proven to be invariant under time reparametrization; See Lemma 2.3 in Lou et al. (2024a). This means that the development of a path is unchanged by the speed at which the path is traversed. This invariance arises because the development is defined through integration along the path, which accumulates increments in order but does not depend on how quickly the path is traversed. As a result, only the sequence of values (i.e., the path itself), rather than its parameterization in time, determines the final representation. It is well suited to action recognition, as the prediction of actions should not depend on the speed of the actions or the frame rate. Time-reparametrization invariance of path development may bring massive dimension reduction and enhance the robustness to variation in speed.

**Uniqueness of the path development** Intuitively, the path development effectively captures the order of events in time series, leveraging the non-commutative nature of matrix multiplication. Changing the order of events may lead to different development in general, highlighting its capacity to capture the temporal dynamics of sequential data. It is proven in (Chevyrev & Lyons, 2016) the path development is a characteristic and universal feature of an un-parameterized path in $V_1([0,T], \mathbb{R}^d)$ under the appropriate Lie group. In other words, transforming a path of finite length to its development map $M \mapsto \text{Dev}_M(X)$ can uniquely determine the path $X$ up to translation. Thus, the path development combined with the starting point effectively summarizes the path without any loss of information.

### 3.2   Path Development Layer

Proposed in  (Lou et al., 2024a), the path development layer is a novel neural network inspired by the path development by exploiting the representation of the matrix Lie group. More specifically, this layer utilizes the parameterized linear map $M \in L(\mathbb{R}^d, \mathfrak{g})$ by its linear coefficients $\theta \in \mathfrak{g}^d$.

**Definition 3.2 (Path development layer)** *The path development layer, denoted by $D_\theta$, is defined as a mapping $\mathbb{R}^{d \times (N+1)} \to G$, which transforms any input time series $x = (x_1, x_2, ..., x_N)$ to the development $z_N \in G$ under the trainable linear map $M_\theta \in L(\mathbb{R}^d, \mathfrak{g})$. For $n \in \{0, ..., N-1\}$, we set*

$$z_{n+1} = z_n \exp(M_\theta(x_{n+1} - x_n)), z_0 = Id_m, \tag{4}$$

*where $\theta \in \mathfrak{g}^d$ is the model parameter.*

Eq. (4) is a direct consequence of Example 3.1 and the multiplicative property of the development (Eq. (3)). Moreover, Eq. (4) shows the recurrence as an analogy to that of the RNNs, albeit in a much simpler manner, as depicted in Figure 1 (lower panel). Similar to RNNs, the path development can cope with time series input of variable length.

**Dimension reduction** The output of the path development has dimension $m^2$, where $m$ is the matrix size. Note that its dimension does not depend on the path dimension $d$ and time dimension $T$, resulting in the massive dimension reduction. This is in stark contrast to the signature feature of degree $k$, whose dimension $\left( \sum_{i=0}^{k} d^i = \frac{d^{k+1}-1}{d-1} \right)$ grows geometrically w.r.t $d$. Moreover, the path development is well suited to extract information from high-frequency time series or long time series without concern on the curse of dimensionality.

**Robustness to irregular sampling** As the path development can be interpreted as a solution to the differential equation (Eq. (1)), it is a continuous time series model. Similar to other continuous time models such as signature and Neural CDEs (Kidger et al., 2020), the path development exhibits robustness in handling time series data with irregular sampling.

**Computational Efficiency of analyzing online data** Multiplicative property of the path development allows incremental computation of the development layer in streaming data, thereby reducing both computational load and memory requirements.

### 3.3   G-Dev Layer: Apply path development layer to a temporal graph

In this subsection, we propose the *G-Dev layer*, which extracts the local information of the temporal graph using the development layer in a rolling window fashion. As a sequence-based extension of the development layer, it generalizes the mapping from $d$-dimensional time series to temporal graph data. Concretely, the G-Dev layer operates over sliding windows along the time dimension, where the application of path development on each local window is referred to as the *G-Dev module*.

Throughout this paper, the temporal graph $\mathfrak{G}_X = (\mathcal{V}, \mathcal{E}, X)$ refers to the graph with each node associated with $d$-dimensional time series features $X$. Here $\mathcal{V} := \{v^1, \cdots, v^M\}$ denotes the vertex set and $\mathcal{E}$ can be represented by the adjacent matrix $A \in \mathbb{R}^{M \times M}$. We consider the general case where the time dimension of each node may vary across the nodes. More specifically, Let $X^i = (x_{t_1}, \cdots, x_{t_{L_i}}) \in \mathbb{R}^{d \times L_i}$ denote the features of the $i^{th}$ vertex, where $d$ and $L_i$ are the path and time dimension, respectively. The feature sets $X = (X^{(1)}, \cdots, X^{(M)})$ is obtained by stacking $X^i$ for all $i \in \{1, \cdots, M\}$. Clearly, the skeleton sequence is an example of temporal graph data.

**G-Dev module**. Let us first define the G-Dev module, which is a trainable layer with model parameters $\theta \in \mathfrak{g}^d \subset \mathbb{R}^{m^2 \times d}$, where $m$ is the matrix order of development. It maps a temporal graph $\mathfrak{G}_X = (\mathcal{V}, \mathcal{E}, X)$ to a static graph with matrix-valued features $\mathfrak{G}_Z = (\mathcal{V}, \mathcal{E}, Z)$ by applying the path development to the time-augmented transformation of each feature vector $X_i$ of the vertex $v_i$, as shown in Figure  1. In formula, for each vertex index $i \in \{1, \cdots, M\}$, the output feature is $Z = (Z^{(i)})_{i=1}^M$, where $Z^{(i)} = D_\theta(X^{(i)}) \in G \subset \mathbb{R}^{m \times m}$.

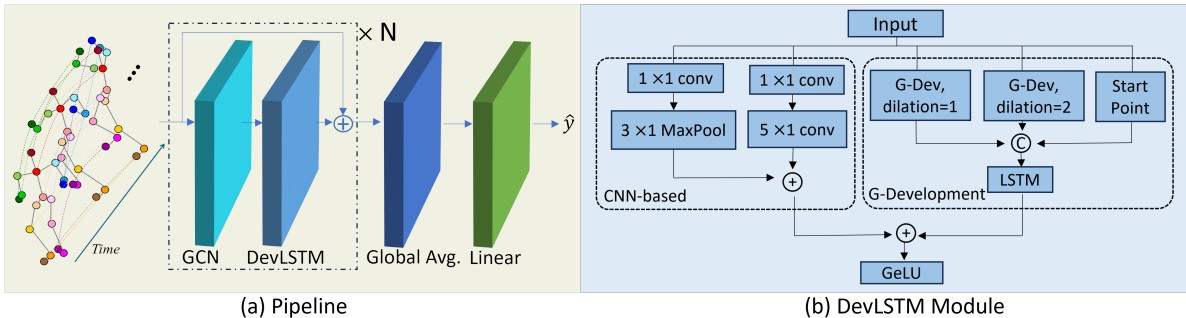

Figure 2: (a) The pipeline of our proposed approach consists of $N$ blocks, with each block containing a GCN module and a DevLSTM module. (b) The detail of the DevLSTM module.

Built on top of the G-Dev module, we employ a sliding window strategy to construct the following G-Dev layer, which can summarize (potentially long) time series over coarse time partitions to reduce the time dimension effectively while retaining the fine-grained information by using the low-dimensional representation.

Without loss of generality, let us focus on the case where the graph features $X$ share the same time stamps $(t_i)_{i=0}^{L-1}$ across vertices[1]. Fix the window (kernel) size $K$ and stride $S$. Let $L' = \lceil \frac{L-k}{S} \rceil$. By applying padding to $Y$, for any $Y \in \mathbb{R}^{d \times N}$, we define a sequence of time series $\tilde{Y}_j$:

$$\tilde{Y}_j = (Y_{j \times S}, \cdots Y_{j \times S + K - 1}) \in \mathbb{R}^{d \times K}, \forall j \in \{0, \cdots, L' - 1\}. \tag{5}$$

**Definition 3.3 (G-Dev layer)** *A G-Dev layer with the matrix degree $m$, window size $K$ and stride $S$ is defined as a trainable map that takes a temporal graph $\mathcal{G}_X = (\mathcal{V}, \mathcal{E}, X)$ with $X \in \mathbb{R}^{M \times L' \times d}$ as an input and outputs a temporal graph $\mathcal{G}_Z = (\mathcal{V}, \mathcal{E}, Z)$ with model parameter $\theta \in \mathbb{R}^{d \times m \times m}$. Here $Z = (z_j)_{j=1}^{L'}$ is a time series consisting of $z_j \in \mathbb{R}^{M \times m \times m}$. Each $z_j = G\text{-}Dev_\theta(\tilde{X}_j)$, where $\tilde{X}_j$ is obtained by applying sliding window by Eq. (5).*

The G-Dev layer (Definition 3.3) can be easily extended to the moving window with the dilation parameter $D$, where we collect the path information at every $D$ time steps during each moving window. To guarantee no information loss of the G-Dev layer, it is crucial to choose $S$ and $K$ to ensure a minimum one-point overlap between two consecutive moving windows.

## 4 GCN-DevLSTM Network for SAR

A skeleton sequence can be modeled by a temporal graph, where the vertex set corresponds to $M$ possible joints, with each $v^i \in \mathbb{R}^2$ or $\mathbb{R}^3$ indicating 2 or $3D$ coordinates of the $i$th joint, respectively. The edge set $\mathcal{E}$ denotes the set of all the joint pairs connected by a bone.

In Figure 2(a), we introduce the proposed GCN-DevLSTM Network, a novel framework for SAR tasks that integrates the GCN and DevLSTM modules into a single block. The GCN module is designed to detect spatial correlations among vertices, while the DevLSTM module handles the temporal dynamics across video frames. Further details on these modules are discussed in Sections 4.1 and 4.2, respectively. Our network is designed to capture discriminative spatial-temporal features with $N$ such blocks, where $N = 9$ in this paper. Additionally, within each block, we build a residual path that establishes a shortcut connection with the input to mitigate gradient vanishing/ explosion issues, enhancing the training stability.

Following these GCN-DevLSTM blocks, we employ global average pooling to effectively reduce both temporal and spatial dimensions to 1. After that, a linear function is applied to adjust the feature dimension to match the number of action classes, thereby facilitating the classification process. Cross-entropy loss is employed at the end to compare the predicted action class with the ground truth.

---

[1]Otherwise, we apply linear interpolation for data imputation, which would not affect the development feature thanks to invariance to time re-parameterization.

### 4.1 GCN Module

In the GCN module, we basically follow the network of CTR-GC (Chen et al., 2021). However, we depart from the use of three initial adjacency matrices representing self, inward, and outward correlations. Instead, we propose a Hops-CTRGC to enhance the model by incorporating various hops of information to construct three adjacency matrices denoted as $A_0$, $A_1$, and $A_2$.

$A_0$ remains an identical (self) matrix. For $A_1$, the elements $a_{ij}$ are set to 1 if vertex i and vertex j are physically connected in the skeleton graph; otherwise, $a_{ij}$ in $A_1$ is set to 0. $A_1$ captures the direct connections in the graph. To enrich spatial information, we introduce $A_2 = A_1^2$, enabling a vertex to connect with its neighbor's neighbor. $A_2$ allows us to extend our understanding beyond direct connections to include relationships involving two hops in the graph. Since GCN module is not our main focus in this paper, readers can find more details about the network of CTR-GC in (Chen et al., 2021) or in our appendix A.

### 4.2 DevLSTM Module

The proposed DevLSTM module, illustrated in Figure 2(b) is mainly composed of the CNN-based branch and the G-development-based branch. The output of DevLSTM module is given by applying GeLU activation function to the summation of the output of both branches.

**G-Development branch**. Inspired by multi-scale TCN (Liu et al., 2020), we also design a multi-scale network to capture temporal dynamics. We utilize two G-dev layers with the dilation parameter $D = 1$ and $D = 2$ to enhance its ability to capture temporal patterns across different scales in the input sequence.

The path development combined with the starting point can determine the path uniquely. Hence, we append the starting point location to the output of the path development layer along the feature dimension over each sub-time interval. The resulting features are then fed into an LSTM network to aggregate local development summaries and capture long-term dependencies in time series data. We use the sequential output of LSTM, preserving the same time dimension as the input.

**CNN-based branch**. MaxPool and a convolutional layer with $5 \times 1$ kernel size have consistently demonstrated superior performance within the temporal module of other GCN-based studies. Introducing both MaxPool and the $5 \times 1$ convolutional operation to our DevLSTM temporal module has the potential to enhance model performance even further. We can reasonably design kernel size, padding, stride and dilation to ensure the output size of Maxpool, LSTM and $5 \times 1$ convolution remain the same.

## 5 Numerical Experiments

To validate the effectiveness of the proposed approach, we carry out numerical experiments on the following popular benchmarking datasets for skeleton-based action and gesture recognition. These datasets are chosen for their diversity in data size and task complexity, providing a comprehensive evaluation of the model's performance. (1) **Chalearn2013** (Escalera et al., 2013) is a public dataset for gesture recognition, consisting of 11,116 skeleton examples and 20 gesture categories, filmed by 27 people. Each frame has 20 joints. (2) **NTU RGB+D 60** (NTU-60) (Shahroudy et al., 2016) consists of 56,880 action samples and 60 action classes, filmed by 40 people using three Kinect cameras in different camera views. 3D skeletal data in the dataset have different sequence lengths, with each owning 25 joints. (3) **NTU RGB+D 120** (NTU-120) (Liu et al., 2019) extends upon NTU RGB+D 60, consisting of 114,480 samples and 120 action classes, filmed by 106 people in three camera views. We consider the Cross-subject (X-Sub) and Cross-view (X-View) evaluation benchmarks for both NTU-60 and NTU-120 as proposed by Shahroudy et al. (2016). We use only skeleton data from these datasets for our experiments.

Experiments are conducted using a single Nvidia A6000 graphics card, where a batch of 16 examples is fed to the network. Stochastic gradient descent optimiser (Bottou, 2012) is employed, with a Nesterov momentum = 0.9 and a weight decay = 0.0003. The training epoch is set to 65 and we apply the warm-up strategy (He et al., 2016) to the first 5 epochs. The initial learning rate is 0.02 and a step learning rate decay is applied with a factor of 0.1 at epoch 35 and 55, respectively. Data processing on NTU datasets

mainly follows the procedure in (Chen et al., 2021), with the length of all input data sequences resizing to 64. We use $\mathfrak{se}$ Lie algebra in this paper, with a corresponding matrix size of $m = 10$. A detailed network architecture can be found in the supplementary. The replicate padding is used in our experiments.

## 5.1 Fusion results from different data streams

Most of the recent approaches that we are set to compare combine predicted scores from different input data streams for enhanced performance. The most commonly used data streams are joint (J), bone (B), joint motion (JM) and bone motion (BM) where details can be found in (Shi et al., 2019; 2020). Apart from the above-mentioned data streams, we introduce a novel stream: the line graph. In a line graph, vertices represent bones, and edges establish connections between these bones (Harary & Norman, 1960). More details about line graph can be found at Appendix B.

## 5.2 Comparison with state-of-the-art methods

Table 1: Results on Chalearn2013 dataset.

|  | Acc (%) |
|---|---|
| Two-streamLSTM (Wang & Wang, 2017) | 91.70 |
| ST-LSTM + Trust Gate (Liu et al., 2018) | 92.00 |
| Multi-path CNN (Liao et al., 2019) | 93.13 |
| 3s_net_TTM (Li et al., 2019a) | 92.08 |
| GCN-Logsig-RNN (Liao et al., 2021) | 92.86 |
| ST-PSM+L-PSM (Cheng et al., 2023) | 94.18 |
| AGGP (Shi et al., 2024) | 95.18 |
| Our method | **95.61** |

Table 1 shows the comparison results between our result and other state-of-the-art methods on Chalearn2013 dataset. We only employ the joint stream for fair comparison as other approaches only use the joint representation on Chalearn2013 dataset. It is evident that our model yields the best result on the small-scale gesture dataset, surpassing the second-best method by 0.42%, proving its effectiveness on the small dataset.

We compare our approach with other state-of-the-art methods on both NTU-60 and NTU-120 datasets, as shown in Table 2. Since our GCN module is following CTR-GCN (Chen et al., 2021) with the major difference lying in the temporal module, we show more results on CTR-GCN for a more comprehensive comparison. As shown in Table 2, all of our configurations, whether using only the joint stream or multi-stream fusion (e.g., joint and bone), consistently outperform CTR-GCN across all benchmarks. Our method's relatively limited performance gain from additional motion streams, compared to CTR-GCN, may stem from the path development layer focusing on position differences between consecutive time steps. As a result, motion streams provide overlapping information at early stages of the network, limiting their contribution. Compared to more recent methods such as BlockGCN (Zhou et al., 2024), which achieves stronger performance when using multi-stream inputs, our method attains comparable results under the joint-stream setting on NTU-60 and remains competitive on the more challenging NTU-120 benchmark.

Overall, our model, integrating multiple data streams, achieves competitive performance across benchmarks while maintaining a consistent network structure across datasets, demonstrating strong adaptability to different dataset sizes. Moreover, it is worth noting that methods such as BlockGCN primarily focus on improving spatial modeling, whereas our work targets temporal modeling, making the two approaches complementary.

## 5.3 Comparison with signature-based models

Since path development layer can be seen as a data-adaptive alternative to the signature, we further compare our method with other signature-based models in SAR task. Compared to the Logsig-RNN (Liao et al., 2021), one of the recent signature-based action recognition methods (using the joint stream only), our model (J) outperforms 10.4% and 9.7% respectively on both benchmarks of NTU-120 dataset, and 2.75% on

Table 2: Experimental results on NTU-60 and NTU-120 dataset. The best results for each benchmark are bolded.

| | | | | | | NTU-60 | | NTU-120 | |
|---|---|---|---|---|---|---|---|---|---|
| | | | | | | X-sub (%) | X-view (%) | X-sub (%) | X-view (%) |
| ST-GCN (Yan et al., 2018) | | | | | | 81.5 | 88.3 | 70.7 | 73.2 |
| AGC-LSTM (Si et al., 2019) | | | | | | 89.2 | 95.0 | - | - |
| 2S-AGCN (Shi et al., 2019) | | | | | | 88.5 | 95.1 | 82.5 | 84.2 |
| Shift-GCN (Cheng et al., 2020b) | | | | | | 90.7 | 96.5 | 85.9 | 87.6 |
| Logsig-RNN (Liao et al., 2021) | | | | | | - | - | 75.8 | 78.0 |
| CTR-GCN(J) (Chen et al., 2021) | | | | | | 89.8 | 94.8 | 84.9 | 86.7 |
| CTR-GCN(J+B) (Chen et al., 2021) | | | | | | 92.0 | 96.3 | 88.7 | 90.1 |
| CTR-GCN (Chen et al., 2021) | | | | | | 92.4 | 96.8 | 88.9 | 90.6 |
| EfficientGCN (Song et al., 2022) | | | | | | 92.1 | 96.1 | 88.7 | 88.9 |
| Ta-CNN (Xu et al., 2022) | | | | | | 90.4 | 94.8 | 85.4 | 86.8 |
| FR-Head (Zhou et al., 2023) | | | | | | 92.8 | 96.8 | 89.5 | 90.9 |
| SPIANet (Yin et al., 2024) | | | | | | 92.8 | 96.8 | 89.2 | 90.4 |
| BlockGCN(J) (Zhou et al., 2024) | | | | | | 90.9 | 95.4 | 86.9 | 88.2 |
| BlockGCN (Zhou et al., 2024) | | | | | | **93.1** | **97.0** | **90.3** | **91.5** |
| FD-GCN (Huo et al., 2026) | | | | | | 92.8 | 96.7 | 89.4 | 90.7 |
| | Joint | Bone | Line | Joint motion | Bone motion | | | | |
| | ✓ | | | | | 90.8 | 95.7 | 86.2 | 87.7 |
| | | ✓ | | | | 91.1 | 95.2 | 87.0 | 88.6 |
| Our methods | | | ✓ | | | 91.2 | 95.3 | 87.1 | 88.7 |
| | ✓ | ✓ | | | | 92.4 | 96.4 | 89.0 | 90.4 |
| | ✓ | ✓ | | ✓ | ✓ | 92.6 | 96.6 | 89.3 | 90.7 |
| | ✓ | ✓ | ✓ | ✓ | ✓ | 92.9 | 96.7 | 89.7 | 91.2 |

Chalearn2013 dataset. Compared to the latest signature-based action recognition approach (Cheng et al., 2023), we still outperform 1.43% on Chalearn2013 dataset while unfortunately, they didn't report their results on NTU dataset, leaving no way to compare. These results support the advantages of using path development that we claimed before, compared with path signature.

## 5.4 Robustness analysis

We comprehensively compare the robustness of our full model (GCN-DevLSTM), CTRGCN (Chen et al., 2021) and our model without path development layer (GCN-LSTM) in terms of missing frames, missing joints and variable sequence length. To this end, we firstly create a new test dataset by randomly dropping or inserting a certain number of frames out of 64 frames to test the robustness of missing frames and variable length. Figure 3a illustrates that our full model with the path development layer consistently outperforms the other methods in terms of accuracy, regardless of the proportion of frames missing or inserting, proving its robustness to irregular sampling and variable sequence length. Notably, at a 47% frame-drop rate, the

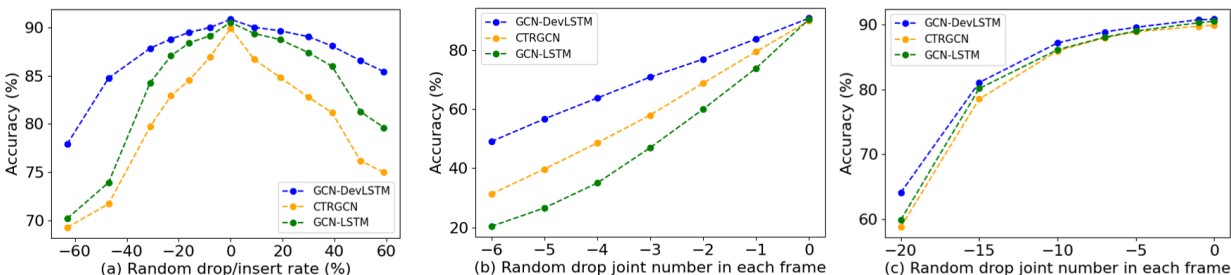

Figure 3: Robustness analysis on NTU60 X-sub benchmark. Figure 3a shows the results of randomly dropping or inserting a certain frames to original time series. Figure 3b and 3c shows the results of randomly dropping a number of joints in each frame with zero-filling and linear interpolation respectively.

accuracy of our full model decreases by only 6.07%, while CTR-GCN drops by 18.18% and GCN-LSTM by 16.56%. A similar trend is observed when frames are inserted.

Additionally, to validate the robustness of missing joints due to occlusions in real life, we create another dataset by randomly dropping a certain number of joints in each frame. To fill in these missing joints, two techniques are adopted: zero-filling (in Figure 3b) and linear interpolation (in Figure 3c). Both figures show that our model with G-Dev layer, is consistently more robust against missing joints than others. More details are in the supplementary material. We credit the superior robustness of our full model to the theory foundation provided by the path development layer, mentioned in 3.2.

## 5.5 Comparison with MS-TCN

To demonstrate the flexible integration of our DevLSTM temporal module with various GCN modules in a plug-and-play fashion, we employ the adaptive graph convolution (AGC) graph in 2S-AGCN (Shi et al., 2019) and a fixed graph without learnable parameters, respectively as an additional GCN module backbone in our network. In addition, we compare our DevLSTM with multi-scale temporal convolution network (MS-TCN) (Chen et al., 2021), commonly used as the temporal module in other recent works across the different GCN backbones. Table 3 illustrates that the proposed DevLSTM module consistently outperforms MS-TCN with different GCN backbones by a large margin, indicating that our designed temporal module is effective in enhancing temporal modeling across various GCN backbones. An overview of adaptive graph convolution and fixed graph can be found in the supplementary material.

Table 3: Comparison between DevLSTM and MS-TCN across various GCN modules on the NTU60 dataset.

| GCN Module | Temporal module | X-sub (%) | X-view (%) |
|---|---|---|---|
| CTR-GC | MS-TCN (J) | 90.3 | 94.9 |
| | DevLSTM (J) | **90.8** | **95.7** |
| AGC | MS-TCN (J) | 89.8 | 94.5 |
| | DevLSTM (J) | **90.6** | **95.6** |
| Fixed graph | MS-TCN (J) | 88.7 | 93.1 |
| | DevLSTM (J) | **89.5** | **94.5** |

## 5.6 Ablation studies

Table 4: Ablation studies of DevLSTM module on NTU-60 dataset. w/o means without.

| Components | X-sub (%) | X-view (%) |
|---|---|---|
| with all components (J) | 90.8 | 95.7 |
| w/o residual path (J) | 89.9 | 95.2 |
| w/o 1× 1 conv & 5×1 conv (J) | 90.7 | 95.3 |
| w/o 1× 1 conv & 3× 1 maxpool (J) | 90.6 | 95.4 |
| w/o G-Dev (J) | 90.5 | 95.2 |

We further investigate the importance of residual path design to connect the input and output of each block and each component in our temporal module. As evident from Table 4, residual path plays an important role to enhance the model performance since it can help alleviate gradient vanishing/explosion in the deep neural network. For temporal components in our DevLSTM module, the path development layer emerges as the primary contributor to the overall performance improvement since removing it causes the largest performance degradation, with other components also contributing to enhanced performance. Importantly, the improvement brought by the G-Dev layer is consistent across both X-sub (+0.3%) and X-view (+0.5%) settings, suggesting that it provides a stable and reliable enhancement rather than a dataset-specific gain.

As further demonstrated in the robustness analysis (Section 5.4), the advantage of the G-Dev layer becomes significantly more pronounced under challenging conditions such as missing frames, missing joints, and

variable sequence lengths. These results suggest that the G-Dev layer's main contribution is not merely improving standard benchmark accuracy, but enhancing robustness and temporal consistency in more realistic and imperfect settings.

Table 5 further shows that the G-Dev layer consistently improves performance across different numbers of blocks, with more pronounced gains in smaller models (e.g., a 3.3% improvement with a single block). As model capacity increases and performance approaches saturation, the marginal gains naturally diminish. Nevertheless, the improvement remains consistent across all configurations.

Table 5: Effect of the G-Dev layer across different numbers of blocks on the NTU-60 X-view dataset. Each block consists of one GCN module followed by one DevLSTM module. All values are reported in %.

| Number of blocks | 1 | 2 | 3 | 4 | 5 | 6 | 7 | 8 | 9 |
|---|---|---|---|---|---|---|---|---|---|
| **With G-Dev layer** | 89.1 | 92.7 | 93.7 | 94.2 | 95.0 | 95.1 | 95.4 | 95.5 | 95.7 |
| **w/o G-Dev layer** | 85.8 | 90.8 | 92.3 | 93.5 | 94.2 | 94.4 | 94.7 | 94.9 | 95.2 |
| **Performance drop** | 3.3 | 1.9 | 1.4 | 0.7 | 0.8 | 0.7 | 0.7 | 0.6 | 0.5 |

## 5.7 Model complexity analysis

Table 6 demonstrates the trade-off between accuracy and model complexity for compact GCN-DevLSTM networks with different numbers of blocks. Increasing the number of blocks from 1 to 9 improves the accuracy from 89.1% to 95.7%, while increasing the number of parameters from 0.13M to 4.61M. However, even with fewer blocks, our results are competitive with much less model parameters in comparison with other sota methods. It is worth noting that our model with 5 blocks only has 68% of model parameters of the strong baseline CTRGCN, reducing 0.49 million parameters (from 1.46M to 0.98M) while slightly outperforming it. Besides, our model with a single block surpasses ST-GCN, despite the differences in preprocessing or training strategies between ST-GCN and our method. Inference time is measured by processing the entire test set with a batch size of 256 and dividing the total runtime by the number of test samples. The slower inference speed compared to CTR-GCN and 2s-AGCN mainly stems from the sequential nature of LSTM and the current implementation of the G-Dev layer. Specifically, LSTM limits temporal parallelization due to recurrent hidden-state updates, while the G-Dev layer requires matrix exponential computations that currently lack optimized CUDA kernel support in common deep learning frameworks. Therefore, the additional runtime overhead is largely implementation-related rather than an intrinsic limitation of path development, and may decrease as GPU support for Lie group operations improves.

Table 6: Model complexity analysis. Compare the model parameters between our approach with the corresponding accuracy on NTU-60 X-view dataset using joint stream only.

| | ST-GCN (Yan et al., 2018) | 2s-AGCN (Shi et al., 2019) | CTRGCN (Chen et al., 2021) | Ours | | | | | | | | |
|---|---|---|---|---|---|---|---|---|---|---|---|---|
| Blocks | - | - | - | 9 | 8 | 7 | 6 | 5 | 4 | 3 | 2 | 1 |
| Params (M) | 1.22 | 1.55 | 1.46 | 4.61 | 3.29 | 1.78 | 1.38 | 0.98 | 0.53 | 0.40 | 0.26 | 0.13 |
| Acc (%) | 88.3 | 94.0 | 94.8 | 95.7 | 95.5 | 95.4 | 95.1 | 95.0 | 94.2 | 93.7 | 92.7 | 89.1 |
| Inference (ms/sample) | - | 1.7 | 1.7 | 11.2 | 10.2 | 8.9 | 8.3 | 7.1 | 5.4 | 4.1 | 2.8 | 1.4 |

## 5.8 Lie Group & Hidden Size Selection

Table 7: Evaluation of accuracy (%) across different hidden sizes in special Euclidean (SE) and special orthogonal (SO) Lie group on NTU60-Xview benchmark with the joint stream.

| | 8 | 9 | 10 | 11 | 12 |
|---|---|---|---|---|---|
| SE | 95.6 | 95.4 | 95.7 | 95.5 | 95.5 |
| SO | 95.5 | 95.5 | 95.4 | 95.6 | 95.4 |

Given that path development is key to our methodology, the choice of a suitable Lie Group, along with its corresponding hidden size, influences the ultimate performance of the model. Table 7 presents results for two commonly used Lie groups with varying hidden sizes. The results indicate that a larger hidden size does not necessarily ensure an enhancement in the model and the best result is obtained by utilizing the

special Euclidean group with a hidden size of 10. These important hyperparameters should be tuned based on numerical results in practice.

## 6 Conclusion and future work

In summary, we propose a novel and generic GCN-DevLSTM network for analyzing temporal graph data via the path development layer. This approach not only offers flexibility when combined with various GCN modules but also significantly enhances performance by effectively capturing the temporal dependencies within data. The effectiveness of the proposed method is validated by the competitive results on three datasets in SAR. Moreover, it demonstrates strong robustness to missing data and variable frame rates.

The current G-Dev layer relies on other GCNs to extract spatial information, which limits its standalone utility. Future enhancements could include integrating the features of adjacent nodes directly into the input of the path development layer to better capture spatio-temporal dependencies.

Our proposed G-Dev layer provides a generic model for temporal graph data, opening up possibilities for its applications across various fields, including urban planning, social network analysis and others.

## Acknowledgements

LJ, WY and HN are supported by the EPSRC [grant number EP/S026347/1]. WY and HN are also supported by The Alan Turing Institute under the EPSRC grant EP/N510129/1. WY and HN are supported by the EPSRC Program Grant [Grant No. UKRI1010] entitled "High order mathematical and computational infrastructure for streamed data that enhance contemporary generative and large language models".

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

## A   Hops-CTRGC

Given three adjacency matrices $A_0, A_1, A_2$ based on the hops representation, following CTR-GC (Chen et al., 2021), we utilize a network to dynamically learn an adjacency matrix for each channel, adding it with $A_k$ for $k \in \{0, 1, 2\}$. This process can be expressed as:

$$A_k^{'}(X) = A_k + \mathcal{F}_\theta(X) \tag{6}$$

where $A_k^{'}$ represents the updated adjacency matrix, $A_k$ is the initial adjacency matrix for the $k$-th hop, and $\mathcal{F}_\theta(X) \in \mathbb{R}^{M \times M \times C}$ dynamically captures the spatial correlations of joints from the network, conditioned on the input data $X \in \mathbb{R}^{T \times M \times d}$, considering each channel of the output channels $C$. Details about the CTR-RC network of $\mathcal{F}_\theta(X)$ can be found in (Chen et al., 2021).

After obtaining the input-conditioned adjacency matrix, the graph convolution is further performed as

$$Z_{t,c} = A_k^{'}(X)_c \cdot (X_t \cdot W_s)_c \tag{7}$$

where $Z \in \mathbb{R}^{T \times M \times C}$ is the output of GCN with the feature dimension $C$, and $W_s \in \mathbb{R}^{d \times C}$ denotes a feature transformation filter. We sum the outputs from three CTR-GC modules, each employing a distinct adjacency matrix. Subsequently, we apply batch normalization and ReLU activation function to the summation result before sending it to the DevLSTM module.

We compare our proposed 3-hop graph representation with I-In-out representations (Chen et al., 2021) on NTU60 dataset. As we can see from Table 8, employing the 3-hop graph representation yields better performance compared to I-In-Out representations on X-view benchmark while maintaining equal performance on X-sub benchmark.

Table 8: Comparison between the graph using I-In-Out and our proposed 3-hop representation in the GCN module on NTU60 dataset.

| Graph Representation | X-sub | X-view |
|:---:|:---:|:---:|
| 3 Hops (J) | 90.8 | 95.7 |
| I-In-Out (J) | 90.8 | 95.5 |

## B   Line Graph

Previous approaches (Shi et al., 2019; 2020; Chen et al., 2021; Zhou et al., 2023) have commonly adopted a multi-stream architecture, including joint, bone, joint motion and bone motion, with each contributing to improved performance. Inspired by (Li et al., 2019b) that leverages a line graph (Harary & Norman, 1960) as the model input, we also integrate a line graph to enhance the performance of our model further. The definition of the line graph is outlined below.

Given a pair of connected joints $V_i$ and joint $V_j$ in the original skeleton graph representation, their bone, denoted by $B_{ij}$, is defined as follows:

$$B_{ij} = V_i - V_j, \forall i < j, (i, j) \in \mathcal{A}, \tag{8}$$

where $\mathcal{A}$ is the admissible set of the pairs of joints shown in all the nodes of the right graph in Figure 4.

In the line graph, each bone serves as the vertex of the graph, as depicted in Figure 4. Whether two given bones $B_{i_1, j_1}$ and $B_{i_2, j_2}$ has the edge is determined by whether they have the shared joint, i.e., $1_{i_1 = i_2} \cdot 1_{j_1 = j_2}$, where $1.$ is an indicator function. The feature of each node $B_{i,j}$ is given by its vector value of $B_{i,j}$.

This utilization of a line graph introduces a novel perspective, enhancing the overall performance of our model. It's clear from Table 2 that incorporating line graph into the whole model, the overall accuracy can get further improved, especially on the NTU-120 dataset, with 0.4% and 0.5% improvement on X-sub and X-view benchmarks, respectively, indicating the importance of the proposed line graph.

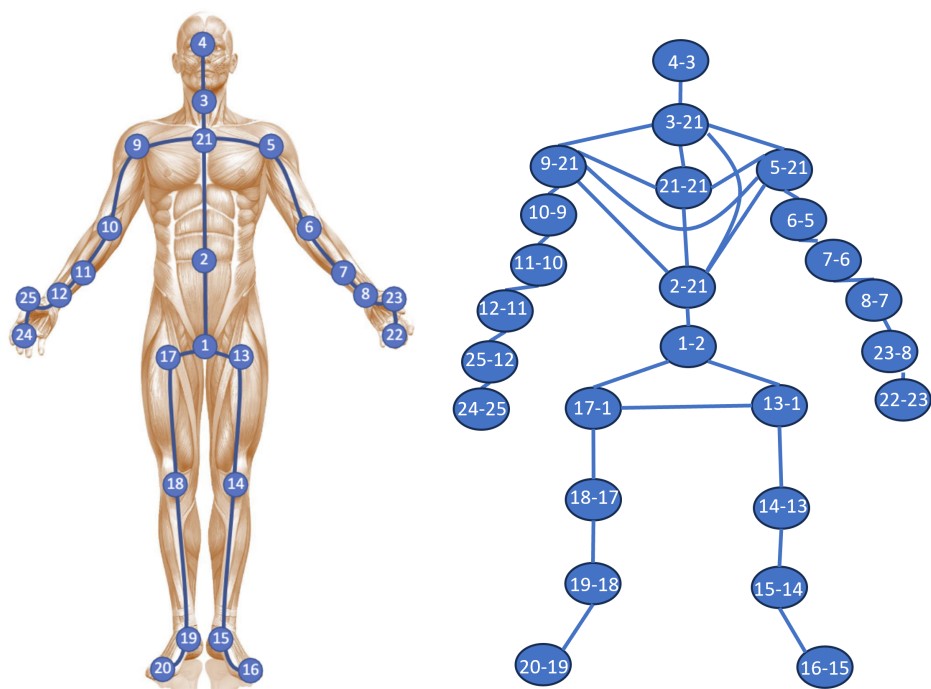

Figure 4: Line Graph. Left side is the original skeleton representation in NTU dataset. The right side is its line graph representation. Joint $V_{1-2}$ in the line graph is the bone $B_{12}$ connecting joint $V_1$ and $V_2$ in original graph.

Additionally, Table 9 investigates the contributions of the G-Dev layer and the line graph under different ensemble settings. The G-Dev layer provides more noticeable improvements in single-stream settings, yielding gains of +0.5% on the joint stream and +0.4% on the bone stream. As more streams are combined, the marginal improvement gradually decreases (+0.2% for Joint+Bone and +0.1% for the four-stream setting), suggesting that multi-stream fusion already captures complementary temporal information. Nevertheless, incorporating the line graph further improves the strongest four-stream model from 96.6% to 96.7%, demonstrating that the line graph provides additional structural information complementary to existing input streams and remains beneficial even in strong multi-stream settings.

Table 9: Ablation study of the contributions of the G-Dev layer and the line graph under different input stream settings. '4s' denotes the four-stream ensemble combining joint, bone, joint motion, and bone motion inputs. Performance gain is calculated as the accuracy improvement brought by the G-Dev layer compared with the corresponding model without the G-Dev layer under the same input stream setting. All values are reported in %.

|            | With G-Dev layer | w/o G-Dev layer | Performance gain |
|------------|------------------|-----------------|------------------|
| Joint      | 95.7             | 95.2            | +0.5             |
| Bone       | 95.2             | 94.8            | +0.4             |
| Joint+Bone | 96.4             | 96.2            | +0.2             |
| 4s         | 96.6             | 96.5            | +0.1             |
| 4s+Line    | 96.7             | 96.7            | 0                |

## C   Comparison with CTR-GCN over parameter number

To further investigate whether the performance improvement arises from the proposed G-Dev layer rather than increased model size, we compare our model with CTR-GCN under the same joint-stream setting across different parameter scales. Specifically, both models are constructed with the same number of blocks, ranging from 1 to 9 blocks, with the 9-block configuration serving as the largest model. To ensure a fair comparison, we further increase the feature dimensions of CTR-GCN while keeping its architecture unchanged, such that its parameter numbers are comparable to those of our corresponding models at each depth.

Figure 5 illustrates the accuracy trend with respect to the number of parameters on the NTU60 X-view benchmark. We observe that our model consistently achieves better accuracy than CTR-GCN under comparable parameter scales, particularly in the low-parameter regime. For example, our 1-block model (0.13M parameters) substantially outperforms the 1-block model of CTR-GCN (0.14M parameters) by 6%, demonstrating the parameter efficiency of the proposed G-Dev layer. As the model size increases, the performance gap gradually narrows due to the saturation effect on NTU60. Nevertheless, our approach maintains a consistent advantage across different scales. These results suggest that the performance improvement mainly comes from enhanced temporal modeling brought by the G-Dev layer rather than simply increasing the model parameters.

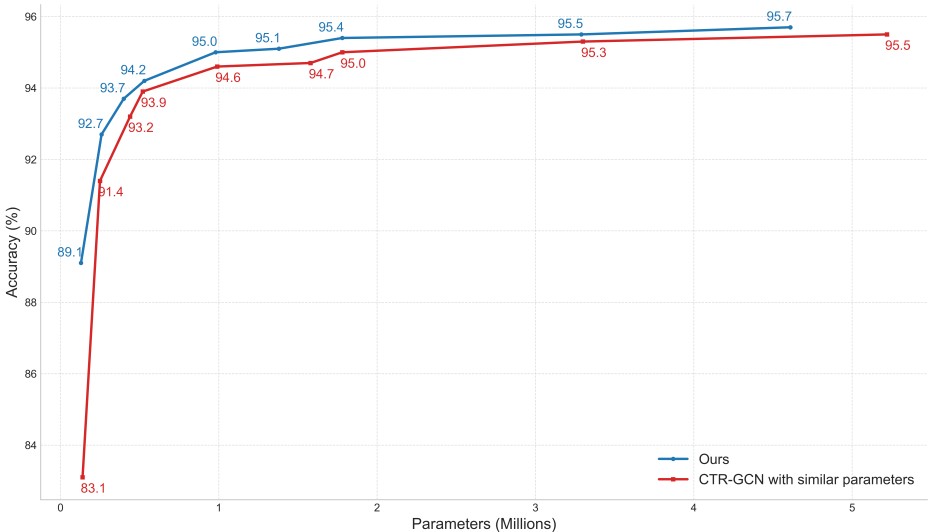

Figure 5: Accuracy trend with respect to the number of parameters on the NTU60 X-view benchmark. Our model consistently achieves higher accuracy than CTR-GCN under comparable parameter scales.

## D   GCN Module

In Figure 6, we present a flow chart illustrating three GCN backbones, serving as the GCN modules employed in this paper for a concise overview. For the details of CTR-GCN and adaptive graph convolution, we refer readers to (Chen et al., 2021) and (Shi et al., 2019), respectively.

The left panel of Figure 6 represents the CTR-GC backbone, while the middle part describes the adaptive graph convolution backbone and the right side is a fixed graph that has no learnable parameters for the adjacency matrix. CTR-GC and adaptive graph convolution share some similarities, as they both utilize three sub-modules and learn the adjacency matrix from the input data. In contrast, CTR-GCN learns the adjacency matrix for each channel, while the channels of adaptive graph convolution share the same adjacency matrix. Besides, adaptive graph convolution utilizes an extra fixed adjacency matrix $B$ representing the physical connection of the body when summing with the learnt adjacency matrix. Different from CTR-GC and adaptive graph convolution, the correlations between joints in the fixed graph are manually defined and remain the same all the time.

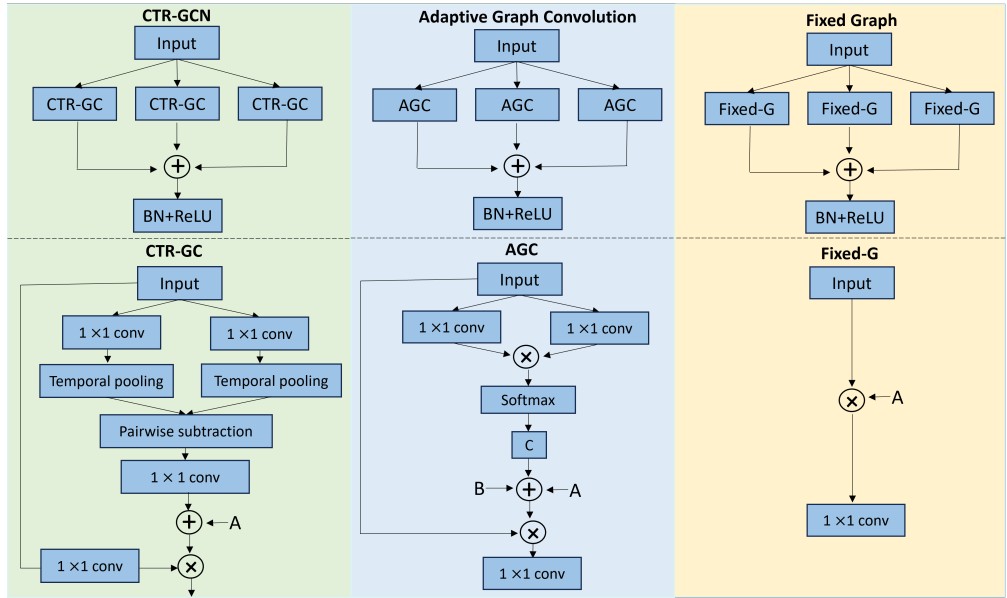

Figure 6: Three GCN Modules used in this paper. The subfigures from left to right represent the CTR-GC module, Adaptive graph convolution module, and the fixed graph, respectively.

# E    Robustness Analysis

We test the robustness of our approach against the missing frames, missing joints and variable sequence legnth on NTU60-Xsub benchmark. There are 64 valid frames at the initial stage and We set up two settings for a comprehensive comparison.

In the first experimental setting, we randomly drop or insert a certain number of frames out of the 64 frames. A corresponding drop or insert rate can be obtained by the drop/insert frames divided by 64. In terms of dropping frame, we use linear interpolation based on the remaining frames to fill in the missing frames while for inserting frames, we directly use the linear interpolation to enlarge the original sequence length. We report the accuracy falling number when increasing drop/insert frames at Table 10. As we can see from Table 10, our full model (GCN-DevLSTM) with path development layer always has much less performance degradation compared to CTRGCN (Chen et al., 2021) and our model without path development layer (GCN-LSTM). The accuracy rate of our full model only drops 6.07% at a dropping rate reaching 47% while CTR-GCN falls 18.18% and GCN-LSTM falls 16.56%. Same situation can also be observed when inserting frame.

Due to occlusion, sometimes not all joints can be captured by cameras, leading to some missing values for these occluded joints. To simulate this situation, in the second robustness setting, instead of dropping the whole frame, we randomly drop a certain number of joints in each frame and fill in the values of missing joints in two different ways:

1. Zero filling: fill missing joint values as zeros as it is a standard way to replace NaN values with zero in machine learning community.

2. Linear interpolation: Estimate the values at intermediate points based on remaining temporal information of each joint. For time points outside the range, we use zeros.

We show the comparison results of both filling ways at Table 11. As we can see that zero filling is more challenging and the performance of all models degrade fast since it will dramatically destroy the consistency of joints along the temporal dimension. But the advantage to use zero filling is that it does not rely on other frames, making it useful when other frames are unavailable. Regardless of filling ways, our full model with

Table 10: Robustness test on NTU60-Xsub benchmark. This table records the accuracy loss when the number of dropping/inserting frame increases.

| Disturbance type | frame number / rate (%) | GCN-DevLSTM | CTRGCN | GCN-LSTM |
|---|---|---|---|---|
| | | Accuracy drop (%) | | |
| Drop | 0 / 0 | 0 | 0 | 0 |
| | 5 / 8 | -0.85 | -3.01 | -1.39 |
| | 10 / 16 | -1.35 | -5.38 | -2.12 |
| | 15 / 23 | -2.04 | -7.01 | -3.46 |
| | 20 / 31 | -2.99 | -10.19 | -6.23 |
| | 30 / 47 | -6.07 | -18.18 | -16.56 |
| | 40 / 63 | -12.93 | -20.65 | -20.31 |
| Insert | 0 / 0 | 0 | 0 | 0 |
| | 6 / 9 | -0.82 | -1.14 | -3.22 |
| | 12 / 19 | -1.18 | -5.09 | -1.77 |
| | 19 / 30 | -1.80 | -7.15 | -3.13 |
| | 25 / 39 | -2.76 | -8.74 | -4.53 |
| | 32 / 50 | -4.27 | -13.73 | -9.23 |
| | 38 / 59 | -5.41 | -14.92 | -10.88 |

Table 11: Robustness test on NTU60-Xsub benchmark. This table records the accuracy loss when the number of dropping joint increases and we use zeros and linear interpolation fillings respectively to those missing joints.

| Filling way | Drop joint number | GCN-DevLSTM | CTRGCN | GCN-LSTM |
|---|---|---|---|---|
| | | Accuracy drop (%) | | |
| Zeros | 0 | 0 | 0 | 0 |
| | 1 | -7.00 | -10.38 | -16.58 |
| | 2 | -13.87 | -21.07 | -30.50 |
| | 3 | -19.92 | -31.91 | -43.50 |
| | 4 | -27.04 | -41.34 | -55.50 |
| | 5 | -34.13 | -50.18 | -63.96 |
| | 6 | -41.76 | -58.65 | -70.16 |
| Linear interpolation | 0 | 0 | 0 | 0 |
| | 1 | -0.09 | -0.18 | -0.22 |
| | 5 | -1.24 | -1.03 | -1.51 |
| | 7 | -1.96 | -1.95 | -2.47 |
| | 10 | -3.65 | -3.96 | -4.38 |
| | 15 | -9.77 | -11.41 | -10.37 |
| | 20 | -26.74 | -31.14 | -30.64 |

path development layer again achieves the strongest robustness against the missing joints compared to other two models.

Our full model with a path development layer demonstrates powerful robustness against the irregular sampling and variable sequence length, benefiting from its theory foundation.

