# OpenReview forum: "GCN-DevLSTM: Path Development for Skeleton-Based Action Recognition"
_TMLR — Accepted by TMLR_

### Review · Reviewer_dfdS · 2026-03-24

**Summary Of Contributions:**

Overall, I think the work is fairly solid. However, the following shortcomings remain:
1 The terminology is somewhat inconsistent. Taking G-Dev as an example, it is referred to as a "layer" in some contexts, while in others it is called a "sequence layer" or "module." This issue significantly affects readability.
2 The so-called dual graph actually has a corresponding definition in graph theory, known as a line graph.
3 The ablation study should place greater emphasis on the contribution of G-Dev, as this is the core innovation of the work.
4 The referenced work is not sufficiently state-of-the-art.

In conclusion, I believe that if the authors take the review comments seriously, the paper has a chance of being accepted.

**Audience:**

Yes

**Audience Explanation:**

In the field of computer vision, skeleton-based action recognition is an important research direction, and solid research results in this area are still meaningful.

**Broader Impact Concerns:**

It is purely technical research with no ethical implications.

**Claims And Evidence:**

Yes

**Claims Explanation:**

The experimental design aligns with its innovations, and the results are credible.

**Requested Changes:**

1 The terminology and symbols throughout the paper need to be reviewed again to ensure consistency.
2 The ablation study section still needs content adjustments, as ablation experiments are a crucial part of verifying the effectiveness of the innovation.
3 It is recommended to add some latest works.

---

> ### Author Response · Authors · 2026-05-13
>
> Thank you for your positive and constructive feedback. We are glad that you find the work solid, and we appreciate your helpful suggestions for improving clarity and positioning. We have revised the manuscript accordingly and address each point below.
>
> ### 1. Terminology inconsistency
> According to your suggestion, we have revised the manuscript to adopt a unified terminology throughout, including Section 3.3 and related descriptions to improve clarity and readability. Specifically, we now consistently use the term **G-Dev layer** to denote the proposed temporal modeling component, replacing the previously used term **G-Dev sequence layer**. Besides, we clarify that the **G-Dev module** denotes the local operation applied within each sliding window, which serves as the fundamental building block of the **G-Dev layer**. Section 3.3 has also been updated to explicitly distinguish the roles of the G-Dev layer and the G-Dev module.
>
> ### 2. Dual graph vs line graph
> We agree that the term *dual graph* is non-standard in this context, and might cause confusion, while the *line graph* is the more accurate terminology used in graph theory. Accordingly, We have replaced *dual graph* with **line graph** in the main text, including the experimental section and ablation studies
>
> ### 3. Ablation study
> We agree that the ablation study should more clearly highlight the contribution of the G-Dev layer.
>
> In the revised manuscript, we have strengthened the analysis in Section 5.6 by adding a more detailed interpretation of the ablation results. Specifically, We further emphasize that the improvement introduced by the G-Dev layer is consistent across both X-sub and X-view settings, indicating a stable and reliable contribution.
>
> In addition, we connect the ablation results with the robustness analysis (Section 5.4), showing that the advantage of the G-Dev layer becomes significantly more pronounced under challenging conditions such as missing frames, missing joints, and variable sequence lengths. We further include additional analysis (Table 5), showing that the G-Dev layer consistently improves performance across different numbers of blocks, with more pronounced gains in low-capacity models (e.g., a 3.3\% improvement with a single block). This further highlights its effectiveness in enhancing temporal modeling.
>
> These revisions provide a clearer and more comprehensive understanding of the contribution of the G-Dev layer.
>
> ### 4. Insufficient coverage of recent SOTA references
> Following the reviewer’s suggestion, we revised Section 2 to improve the coverage of recent state-of-the-art methods and related studies.
>
> Specifically, we have added several recent works, including transformer-based and hybrid GCN–transformer approaches, as well as more recent GCN-based methods. We also replaced or streamlined some older references to ensure that the discussion reflects current developments in skeleton-based action recognition.
>
> In addition, we clarified the positioning of our work with respect to these recent approaches, emphasizing that while many state-of-the-art methods focus on enhancing spatial modeling or leveraging global attention mechanisms, our work addresses the relatively underexplored problem of improving temporal modeling within the GCN framework.

---

> > ### Comment · Reviewer_dfdS · 2026-06-01
> > **This manuscript can be considered for acceptance**
> >
> > The issues and concerns I mentioned have all been addressed by the author. The current version is acceptable to me.

---

### Review · Reviewer_W7wW · 2026-04-03

**Summary Of Contributions:**

The authors argue while it is clear that graph convolutional networks (GCNs) help for learning the spatial relationships for skeletal action recognition (SAR), they say that it is not clear that temporal convolutions are the right way to learn temporal relationships.

The main contributions of the paper are
(a) to extend the Path Development Layer, proposed by Lou et al., 2022 that provides more compact reparameterization-invariant representations of time series, to temporal graphs for the application of SAR.
(b) combining the Path Development Layer with an LSTM for effective SAR.

The experiments on commonly used SAR datasets do show that the proposed architecture outperforms the baselines considered in the paper. The main strength appears to be that the proposed model, GCV-DevLSTM, is more robust to randomly dropping or inserting frames and joints.

Please see the next box for the weaknesses in the paper.

**Audience:**

Yes

**Audience Explanation:**

The paper develops a neural network architecture for skeletal action recognition, which is an important task in computer vision. So, I think a part of TMLR's audience will be interested in this paper.

**Broader Impact Concerns:**

No concerns.

**Claims And Evidence:**

No

**Claims Explanation:**

1. One of the main contributions, according to the authors, is the "G-Dev" layer, which extends the path development layer to graph data. My understanding is that G-Dev just takes the output of the GCN module and applies path development for every channel in the time series independently, before feeding to the LSTM. I don't see this as an extension, it's the same path development layer from Lou et al., 2022 applied to each channel separately. I hope the authors can clarify this point.

2. I appreciate Section 5.6 and Table 4 showing the ablation study. However, it shows that the G-Dev layer has a miniscule effect on the performance for the NTU-60 dataset. I agree that, of all the other components studied, removing the G-Dev layer causes the largest drop, the drop seems to be very small, in my opinion.

3. Related to the above point, as none of the results show statistical significance, it is hard to know whether the proposed method is significantly improving performance. I think the usefulness of the proposed method is obvious only in the robustness analysis.

4. The baselines considered in the paper are not state-of-the-art. For example, please see these two papers that beat the performance shown in this paper:

Zhou et al., BlockGCN: Redefine Topology Awareness for Skeleton-Based Action Recognition, CVPR 2024
Do and Kim, SkateFormer: Skeletal-Temporal Transformer for Human Action Recognition, arXiv 2024

The above points show that some of the claims in the paper are not accurate and the experimental evidence is not convincing in some cases.

**Requested Changes:**

Please address the weaknesses I have mentioned above.

In addition, I think several parts of the presentation of the paper need improvement.

1. Citations. There is an issue with the way the papers are being cited in the paper. Textual citations (\citet) are used everywhere, even though they should be parenthetical (\citep).

2. Some parts of Section 3.1 are likely to be difficult to read for the intended audience. I think it's good to provide a little bit of a higher level view of what the path development formulation is trying to achieve, in simpler terms. There is a mention of how this representation is parameterization-invariant, but I don't see why that is the case.

3. Section 2 has a similarissue. There is a discussion on path signature and path development, but there is no real clarity that the reader can glean from reading it, what is a path signature? I think it's important to explain this more to describe the advantage of path development as an alternative. There is jargon, like "Playing a role as the non-commutative monomials on the path space..", I don't understand this sentence.

I hope the authors can improve the clarity of the paper.

---

> ### Author Response · Authors · 2026-05-13
>
> We thank the reviewer for the detailed and constructive feedback. Below we clarify the key concerns and outline concrete revisions.
> ### 1. Clarification of the G-Dev layer and its novelty
> While the path development operation is applied to each node feature sequence, the novelty of the proposed G-Dev layer lies in extending path development from vector-valued sequences to temporal graph data and integrating it into a unified spatio-temporal framework.
>
> Specifically:
>
> - The G-Dev layer operates on temporal graph data, where each node is associated with a time series, rather than a single vector-valued sequence.
> - It is combined with a sliding-window mechanism to produce a sequence of graph-level representations, enabling temporal downsampling while preserving high-frequency information.
> - It is tightly integrated with GCN and LSTM modules to form a unified spatio-temporal architecture, which differs from prior uses of path development on standalone sequences.
> - In Lou et al. (2022), path development is applied only once after the LSTM. In contrast, we propose a more stable architecture that allows the G-Dev layer to be integrated and stacked within the network, enabling deeper and more expressive temporal modeling.
>
> ### 2. Effect size of the G-Dev layer (Table 4)
> We agree that the gains reported in Table 4 are relatively modest. However, this is common in skeleton-based action recognition, where strong baselines such as CTR-GCN are already highly optimized and improvements are typically incremental.
>
> FD-GCN (Huo et al., 2026), which also builds upon CTR-GCN, reports gains of only +0.1\% on X-Sub and +0.3\% on X-View. Similarly, performance gaps among recent state-of-the-art methods on NTU benchmarks are generally very small. Under the joint-stream setting on NTU-60, BlockGCN exceeds our method by only +0.1\% on X-Sub, while our method achieves +0.3\% higher performance on X-View.
>
> In addition, Table 6 shows that increasing the number of blocks from 7 to 9 substantially increases parameters (1.78M to 4.61M) while improving accuracy by only +0.3\%, illustrating diminishing returns in this regime. Therefore, even small but consistent improvements are meaningful.
>
> Importantly, the advantages of the G-Dev layer become much more pronounced in the robustness analysis (Section 5.4). We also added further analysis in Table 5 showing that removing the G-Dev layer consistently degrades performance across all model scales, with especially large drops in smaller models (e.g., 3.3\% with a single block). This consistent trend suggests that the gains are systematic rather than due to random variation.
> ### 3. Comparison with more recent state-of-the-art methods
> We agree that the original manuscript did not sufficiently reflect the current state-of-the-art landscape. Accordingly, we revised the experimental comparison and related discussion.
>
> First, we explicitly include BlockGCN in Table 2. Under the same joint-stream setting, BlockGCN reports (90.9, 95.4, 86.9, 88.2) on NTU-60 and NTU-120, while our method achieves (90.8, 95.7, 86.2, 87.7). These results show that our method achieves comparable or slightly better performance on NTU-60 while trailing behind on NTU-120. We clarified this comparison in Section 5.2 for a more transparent evaluation.
>
> Second, although our experiments mainly focus on GCN-based methods due to our emphasis on temporal modeling within GCN frameworks, we agree that transformer-based approaches should also be discussed. Therefore, we expanded the related work section to include recent transformer-based methods such as SkateFormer. While transformer-based models focus on capturing global dependencies through attention mechanisms, our work focuses on improving temporal modeling within GCN-based architectures. We view these directions as complementary rather than directly competing.
>
> Finally, we revised the claims throughout the paper to avoid overstating performance. Instead of claiming overall state-of-the-art performance, we now state that our method achieves competitive results while consistently improving strong GCN baselines on NTU datasets.
>
> ### 4. Clarity and presentation
> We carefully reviewed and corrected the citation style throughout the manuscript. Specifically, we now consistently use parenthetical citations (`\citep`) for standard references and reserve textual citations (`\citet`) only when the authors explicitly form part of the sentence.
>
> We also revised Sections 2.2 and 3.1 to improve clarity and accessibility. In Section 2.2, we added a more intuitive explanation of path signature and clarified the motivation for path development as a trainable and compact alternative. We also rephrased overly technical statements to improve readability. In Section 3.1, we added a high-level introduction before the formal definition to explain the purpose of path development more intuitively and clarified the intuition behind time-reparameterization invariance.

---

### Review · Reviewer_QrZL · 2026-05-02

**Summary Of Contributions:**

This work examines the problem of skeleton-based action recognition (SAR) by introducing a new network layer based on path development. In particular, the paths studied in this work are the paths of skeleton joints over time. Path development models a path in Euclidean space by defining a linear mapping of infinitesimal changes in the coordinate space to Lie algebra vectors, and integrating over the Lie algebra vectors associated with the path changes. This map paths to elements of the associated Lie group in a way that encodes characteristics of the path changes. To incorporate this in a neural network layer, the linear mapping from the path tangents to the Lie algebra is treated as a learnable parameter. The proposed network layer has two parts: one is a CNN branch that does pooling and convolution over a short frame range, and the other uses the learnable path development layers to obtain features which are fed into an LSTM. Experiments are conducted on standard action recognition datasets to show that the proposed model can provide strong action recognition performance with a slight edge over leading methods.

**Audience:**

Yes

**Audience Explanation:**

SAR has useful applications for modeling video data, and the skeleton-based approach can be useful for robustness and privacy. The methodology of path development as a neural network feature is interesting and presented clearly.

**Claims And Evidence:**

Yes

**Claims Explanation:**

To some extent, the claims are supported. The ablation of dropping the path development branch (dropping path development leads to a noticeable decrease) and the model size comparison (at the same scale, models with path development are a bit stronger than a pure GCN solution) provide moderate evidence that including the path development layers are important. There seems to be some evidence that the dual data stream could be more influential than the path development layer based on the results in the main table. I am curious how much path development matters when all data streams are used. Further comparisons with the strongest GCN-based model of Chen 2021 and the strongest model of the proposed paper at difference sizes (with the same input streams as the GCN solution) would also elucidate the role of the path development layers more clearly.

**Requested Changes:**

It would be helpful to see the ablation table repeated for the strongest model in this work (to see how much path development still matters when all data streams are included). I also might recommend shifting the paper focus to both path development and the dual data stream. It would also be good to perform a comparison against the strongest GCN model of Chen et. al 2021 and the matching data streams for the proposed model at different sizes. Furthermore, runtime in addition to model size should be presented in the comparisons, since this model has additional layers which might take longer to run at a equal size. The goal of these requested changes is to better isolate the contribution from the path development layers and the dual data stream, and to get an idea of how much performance is coming from scaling, as well as the time cost of additional layers even with the same model size.

---

> ### Author Response · Authors · 2026-05-13
>
> We thank the reviewer for the constructive suggestions regarding the isolation of the contributions of the G-Dev layer, the dual graph (the term has been changed to line graph in the revision), and the effect of model scaling and runtime.
>
> In response, we have added several new experiments and analyses to better clarify these aspects.
>
> First, following the reviewer’s suggestion, we show the ablation results without G-dev layer under stronger multi-stream settings in the newly added Table 9 in Appendix. Specifically, we analyze the contribution of the G-Dev layer under joint, bone, joint+bone, four-stream (4s), and four-stream with line graph settings. The results show that the G-Dev layer consistently improves performance across in single-stream and standard multi-stream settings. In particular, the gain is more pronounced in single-stream settings (+0.5\% on joint and +0.4\% on bone), while the marginal improvement gradually decreases in stronger multi-stream settings (+0.2\% on Joint+Bone and +0.1\% on 4s). This suggests that multi-stream fusion already captures complementary temporal information. Nevertheless, incorporating the line graph further improves the strongest four-stream model from 96.6\% to 96.7\%, demonstrating that the line graph provides additional structural information complementary to existing input streams. Based on this observation, we revised the paper to place greater emphasis on the line graph, highlighting it as one of the main novelties in the introduction.
>
> Second, to better isolate the effect of the proposed G-Dev layer from model scaling, we add a new comparison against CTR-GCN in Appendix C (“Comparison with CTR-GCN over parameter number”). In this experiment, both models are evaluated under the same joint-stream setting across different parameter scales. We vary the number of blocks from 1 to 9 and further adjust the feature dimensions of CTR-GCN such that the parameter numbers are comparable to those of our model. Figure 5 shows that our model consistently achieves higher accuracy than CTR-GCN under comparable parameter scales, particularly in the low-parameter regime. For example, our 1-block model (0.13M parameters) substantially outperforms the 1-block model of CTR-GCN (0.14M parameters) by 6\%. These results suggest that the performance gain mainly arises from the enhanced temporal modeling brought by the G-Dev layer rather than simply increasing model parameters.
>
> Third, following the reviewer’s recommendation, we additionally include runtime analysis in Table 6 alongside parameter counts and accuracy. We further discuss the runtime overhead in Section 5.7. Specifically, we explain that the slower inference speed mainly comes from the recurrent nature of LSTM and the current implementation of the G-Dev layer, particularly the computation of matrix exponentials, which currently lack dedicated fused CUDA kernel support in common deep learning frameworks. We also clarify that this overhead is largely implementation-level rather than an intrinsic limitation of path development itself.
>
> Overall, these additional experiments and analyses provide a more controlled and comprehensive evaluation of the respective contributions of the G-Dev layer, the line graph/dual-stream design, model scaling, and runtime efficiency.

---

### Decision · Action_Editor_T3Hw · 2026-06-22

**Recommendation:** Accept as is

**Audience:**

Yes

**Audience Explanation:**

GCN and action recognition are important topics in modern AI.

**Claims And Evidence:**

Yes

**Claims Explanation:**

This paper proposes the G-Dev layer, which leverages path development to enhance temporal modeling in skeleton-based action recognition, and integrates it with GCNs and LSTMs to achieve competitive performance on NTU-60, NTU-120, and Chalearn2013.

The three reviewers recognized the technical soundness and relevance of the work, but raised concerns regarding the clarity of the G-Dev layer's novelty, insufficient comparisons with recent state-of-the-art methods (e.g., BlockGCN, SkateFormer), modest gains in ablation studies, lack of runtime analysis, and inconsistent terminology. In their rebuttal and revised manuscript, the authors thoroughly addressed these issues by clarifying the extension of path development to temporal graph data, adding new ablation and scaling experiments, including runtime analysis, updating baselines and related work, unifying terminology, and tempering their claims to "competitive performance."

All reviewers confirmed that their concerns were satisfactorily resolved, with two explicitly recommending acceptance and one leaning accept. Based on these recommendations and my own review, I recommend accepting the paper.